# Sensory Neurons Release Cardioprotective Factors in an In Vitro Ischemia Model

**DOI:** 10.3390/biomedicines12081856

**Published:** 2024-08-15

**Authors:** Clara Hoebart, Attila Kiss, Bruno K. Podesser, Ammar Tahir, Michael J. M. Fischer, Stefan Heber

**Affiliations:** 1Institute of Physiology, Center for Physiology and Pharmacology, Medical University of Vienna, 1090 Vienna, Austria; clara.hoebart@meduniwien.ac.at (C.H.); stefan.heber@meduniwien.ac.at (S.H.); 2Center for Biomedical Research and Translational Surgery, Medical University of Vienna, 1090 Vienna, Austria; attila.kiss@meduniwien.ac.at (A.K.); bruno.podesser@meduniwien.ac.at (B.K.P.); 3Division of Pharmacognosy, University of Vienna, 1090 Vienna, Austria; ammar.tahir@univie.ac.at

**Keywords:** sensory neurons, cardiomyocytes, ischemia, ischemia-reperfusion

## Abstract

Sensory neurons densely innervate the myocardium. The role of their sensing and response to acute and prolonged ischemia is largely unclear. In a cellular model of ischemia-reperfusion injury, the presence of sensory neurons increases cardiomyocyte survival. Here, after the exclusion of classical neurotransmitter release, and measurement of cytokine release, we modified the experiment from a direct co-culture of primary murine cardiomyocytes and sensory neurons to a transfer of the supernatant. Sensory neurons were exposed to ischemia and the resulting conditioned supernatant was transferred onto cardiomyocytes. This approach largely increased the tolerance of cardiomyocytes to ischemia and reperfusion. Towards the identification of the mechanism, it was demonstrated that after ten-fold dilution, the conditioned solution lost its protective effect. The effect remained after removal of extracellular vesicles by ultracentrifugation, and was not affected by exposure to protease activity, and fractionation pointed towards a hydrophilic agent. Solutions conditioned by HEK293t cells or 3T3 fibroblasts also increase cardiomyocyte survival, but to a lower degree. A metabolomic search identified 64 at least two-fold changed metabolites and lipids. Many of these could be identified and are involved in essential cellular functions. In the presented model for ischemia-reperfusion, sensory neurons secrete one or more cardioprotective substances that can improve cardiomyocyte survival.

## 1. Introduction

Cardiovascular diseases are the most common cause of death worldwide [1]. Within these, acute myocardial infarction is an important entity that entails the blockade of a coronary artery, which leads to ischemia in the myocardium and cell death of cardiomyocytes [2]. The reperfusion of the occluded vessel by percutaneous coronary intervention is a standard therapy to limit cardiomyocyte cell death [3], but reperfusion per se causes an exacerbation of the myocardial injury, which is called ischemia-reperfusion injury [4]. However, ischemia-reperfusion injury can be targeted by ischemic conditioning and various potentially cardioprotective substances have been tested. During remote pre- or postconditioning, several cycles of ischemia and reperfusion are induced at a remote body site, before or after the beginning of ischemia, respectively. In some studies, also including patients, this then led to a reduction in the infarct size, but overall large clinical trials could not verify a clinically meaningful cardioprotective effect. Additional proposed cardioprotective treatment strategies are acidic reperfusion procedures to slowly readapt cardiomyocytes to physiological pH or targeting reactive oxygen species accumulating in the myocardium. Other proposed targets in cardiomyocytes are ion channels, mitochondria, Ca^2+^ overload, and autophagy. However, all of these targets failed to show clinical efficacy or were only beneficial in a few studies [5,6]. Therefore, the lack of a therapy that targets ischemia-reperfusion injury requires the investigation of novel targets.

Among the ion channels described to be potential targets for cardioprotective effects is also the transient receptor potential channel family, especially the members TRPA1 and TRPV1, for which both protective and detrimental effects have been described in the literature [7]. However, we could show that on the one hand these channels are not functionally expressed in cardiomyocytes and on the other hand targeting TRPA1 with an agonist or antagonist does not reduce the infarct size in rat and mouse models of acute myocardial infarction [8,9]. Nevertheless, sensory neurons derived from primary murine dorsal root ganglia (DRG) markedly increased cardiomyocyte survival when cultured together in ischemia-reperfusion conditions in vitro [9]. With this in mind, the aim of this study was to discover the underlying mechanisms and identify potentially cardioprotective substances.

In in vitro models of ischemia-reperfusion injury, conditions are often mimicked by hypoxia and reoxygenation. As described by the European Society of Cardiology, ischemic injury can be further mimicked more closely by producing a glucose-free, hyperkalemic, and acidic environment that additionally contains metabolic end products like lactate. Reperfusion conditions can then be induced by balancing these factors back to physiologic conditions [10]. This recommended combination of factors was also used in our in vitro ischemia-reperfusion model. As we could show previously, a combination of 2 h of such ischemic conditions and 0.5 h of mimicked reperfusion leads to reduction in cardiomyocyte survival by approximately 50%, which allows us to study factors that are protective and factors that are detrimental for cardiomyocytes [9]. In this study, experiments were conducted to obtain insights into the cardioprotective mechanism using the co-culture model of cardiomyocytes and sensory neurons in one dish. Subsequently, the experimental approach was modified to demonstrate that the previously observed protective mechanism does not rely on direct cell–cell contacts, but primarily depends on something released by sensory neurons into the environment upon ischemic stimulation. Specifically, this was achieved by challenging sensory neurons by ischemia and transferring the supernatant to cardiomyocytes. The objective was to narrow down or identify the protective mediators. As sensory neurons densely innervate the myocardium [11], it is plausible that substances released from neurons would affect cardiomyocytes. Sensory neurons are well known to release transmitters like a calcitonin gene-related peptide, which contributes to local perfusion but also neurogenic inflammation [12]. However, neurons release a variety of other less investigated molecules. This study aimed to identify transferable mediators of cardioprotection.

## 2. Materials and Methods

### 2.1. Chemicals and Solutions

To isolate cardiomyocytes, a Ca^2+^ free solution was used, which contains (in mM) 134 NaCl, 11 glucose, 4 KCl, 1.2 MgSO_4_, 1.2 Na_2_HPO_4_, and 10 HEPES. This solution was buffered to pH 7.35 with NaOH and the myosin inhibitor 2,3-butanedione monoxime (BDM) at 10 mM was added. Above-mentioned salts and HEPES were obtained from Carl Roth (Karlsruhe, Germany), Sigma-Aldrich (St. Louis, MO, USA), or Merck (Darmstadt, Germany), and NaOH from Thermo Fisher Scientific (Waltham, MA, USA). The “ischemic solution” contained (in mM) 137 NaCl, 8 KCl, 0.49 MgCl_2_6H_2_O, 0.9 CaCl_2_2H_2_O, 4 HEPES, and 20 sodium lactate and was set to a pH of 6.2. The “reperfusion solution” contained (in mM) 137 NaCl, 2 KCl, 1.5 MgCl_2_6H_2_O, 0.9 CaCl_2_2H_2_O, 4 HEPES, and 12 glucose and was set to a pH of 8.8. The “external solution” contained (in mM) 145 NaCl, 5 KCl, 1 MgCl_2_6H_2_O, 1.25 CaCl_2_2H_2_O, 10 HEPES, and 10 glucose and was set to a pH of 7.4. KCl, MgCl_2_6H_2_O, and sodium lactate were obtained from Sigma-Aldrich; CaCl_2_2H_2_O was obtained from Merck; NaCl and HEPES were from Carl Roth; and glucose was from Thermo Fisher Scientific. Olcegepant (BIBN4096BS) was obtained from Sigma-Aldrich, and botulinum toxin A from Metabiologics (Madison, WI, USA). Carbogen gas, consisting of 95% N_2_ and 5% CO_2_, was supplied by Linde Gas (Dublin, Ireland) or Messer Group (Bad Soden am Taunus, Germany).

### 2.2. Animals

Breeding, euthanasia, and all procedures of animal handling were performed according to regulations of animal care and welfare. Experiments were carried out in accordance with the European Communities Council Directive of 24 November 1986 (86/609/EEC). Male and female wild-type C57Bl/6J mice starting at 8 weeks of age (Department for Laboratory Animal Science and Genetics, Himberg, Austria) were euthanized by cervical dislocation, preceded by anesthesia by exposure to isoflurane (CP Pharma, Burgdorf, Germany). According to Austrian law, killing animals solely for the purpose of utilizing their tissues or organs does not qualify as an animal experiment and therefore does not require authorization by a committee.

### 2.3. Isolation of Primary Cardiomyocytes

A volume of 5 mL of an ice-cold calcium-free solution was injected into the ventricles to clear them of blood and halt contractions. Subsequently, the heart was surgically excised, suspended by the aorta in the Langendorff apparatus [13], and retrograde-perfused at 37 °C for 3 min with a calcium-free solution. This was followed by an 18 min perfusion at 0.18 mL/min using a calcium-free solution containing 0.17 mg/mL collagenases (Liberase, Roche, Basel, Switzerland). The atria were discarded, and the ventricles were mechanically triturated with fine forceps. The cells were then incubated on a shaker at 37 °C and exposed to solutions with increasing calcium concentrations over one hour, reaching a maximum of 0.2 mM calcium in five steps. After a final mechanical trituration with a plastic pipette, the isolated cardiomyocytes were centrifuged at 9× *g* for 3 min and resuspended using a glass pipette in the cardiomyocyte medium containing M199 (M5017). This medium was supplemented with 10 µg/mL insulin, 5.5 µg/mL transferrin, 5 ng/mL selenium, 0.1% BSA, 10 mmol/l BDM, 1% penicillin/streptomycin, 1% L-glutamine (all Sigma-Aldrich), and a 1X CD lipid (Thermo Fisher Scientific) according to Ackers-Johnson et al. [14]. The cardiomyocytes were then seeded into 24-well plates coated with laminin-functionalized peptigel alpha 2-IKVAV (Manchester Biogel, Manchester, UK). Cardiomyocytes were allowed to recover and to attach to the peptigel for 30 min at 37 °C and 5% CO_2_. Brightfield images were captured using an Olympus IX73 inverted microscope with a 4× objective (Olympus, Tokyo, Japan).

### 2.4. Measurement and Analysis of Cytokines in Cell Culture Supernatants

As previously reported (Hoebart at al., 2023 [9]), cardiomyocytes were either cultured alone or co-cultured with sensory neurons derived from DRG. Cells were kept in control conditions or exposed to conditions mimicking ischemia-reperfusion. These experiments were repeated using 3 animals and then the pooled supernatants of these 3 identical experiments were concentrated using an Amicon Ultra filter (3 kDA, 15 mL, Sigma-Aldrich). Thereafter, the proteome profiler mouse XL cytokine array kit (Biotechne, Minneapolis, MN, USA) was used according to the manufacturer’s instructions. In short, the membranes were first blocked with Array Buffer 6 on a shaker for 1 h and then incubated with the filtrates of the pooled supernatants overnight at 4 °C on a shaker. On the next day, the membranes were washed 3 times, incubated with a detection antibody for 1 h on a shaker, washed 3 times again, and then incubated with streptavidin–horseradish peroxidase for 30 min on a shaker. The blots were then developed with the chemiluminescent reagent mix of the kit and imaged with a Western Blot imager (UVP ChemStudio, Analytik Jena, Jena, Germany). Images were equally contrasted, individual dots analyzed using Fiji [15], and background values subtracted using Excel (Microsoft, Albuquerque, NM, USA). Using GraphPad Prism 9 (Graphpad Software, Inc, San Diego, CA, USA), a heatmap was generated using the log2 of background-subtracted mean values of the technical replicates and showing the fold changes of ischemia vs. control in cardiomyocytes cultured alone and in co-culture. To visualize which analytes were altered between conditions, the background-subtracted values were log transformed and a regression line was fitted. Scatter plots of transformed data with one condition on each axis and a line of identity (y = 0 + 1*x) indicated that some conditions resulted in general shifts in abundance. As this was considered a technical artifact, a regression line was fitted with a fixed slope (y = Intercept + 1*x), allowing only the intercept to vary. Cytokines deviating from the regression line were identified using the ROUT method with a Q of 5% [16]. To find out whether pathways might be differentially regulated between the groups, the raw background-subtracted data of each technical replicate were analyzed using the ExpressAnalyst online tool after log2 transformation [17] and ridgeline plots were generated by performing a GSEA (gene set enrichment analysis) with the rank Welch’s *t*-test in the KEGG (Kyoto Encyclopedia of Genes and Genomes) database. For display in Appendix A, a ridgeline diagram was generated using a tricube kernel in IBM SPSS statistics 28 (Armonk, NY, USA) and assembled with the individual gene’s fold changes in CorelDraw 17 (Corel Corporation, Ottawa, ON, Canada). It shall be stressed that this analysis was purely performed to generate hypotheses and must not be interpreted as confirmatory.

### 2.5. Conditioned Ischemic Solution

DRG from all spinal levels were excised from adult wild-type C57/Bl6J mice, as previously described [18]. Whole DRG were incubated in an ischemic solution and additionally deprived from oxygen by incubation in an anoxic environment in a modular incubator chamber (Embrient, San Diego, CA, USA) flushed with gas containing 95% N_2_ and 5% CO_2_ (Linde or Messer) at 37 °C for 2 h to mimic ischemic conditions. For each independent experiment, DRG neurons from one mouse were used to prepare a “DRG-conditioned ischemic solution”, which was used for one ischemia-reperfusion model experiment on cardiomyocytes.

Ischemia-conditioned supernatants of human embryonic kidney 293t cells (HEK293t) and 3T3 fibroblasts, both obtained from ATCC (Manassas, VA, USA), were generated similarly. These two cell types were chosen as cell types of interest and an example of a commonly used cell line. HEK293t cells were seeded at 12,500 cells/well and 3T3 fibroblasts at 7000 cells/well in a 24-well plate in Dulbecco′s Modified Eagle′s Medium (Sigma-Aldrich) supplemented with 100 mg/mL streptomycin/penicillin (Lonza, Walkersville, MD, USA) and 1% L-glutamine (Lonza). These densities were chosen to cover approximately about the same area as the sensory neurons did in co-culture with cardiomyocytes [9]. As above, the cell lines were then incubated at 37 °C for 2 h to mimic ischemic conditions.

This DRG- or HEK293t-conditioned ischemic solution was also diluted 1:10, 1:100, and 1:1000 in an ischemic solution to generate a dose–response curve. Solutions were either stored in glass vials (Agilent, Santa Clara, CA, USA) at −20 °C or lyophilized (Martin Christ, Osterode am Harz, Germany) before storage at −20 °C.

### 2.6. Removal of Extracellular Vesicles

The DRG-conditioned ischemic solution was centrifuged at 300× *g* for 10 min at room temperature to remove debris from cells and the supernatant was then frozen at −20 °C. This supernatant was centrifuged at 120,000× *g* (Optima, Beckman Coulter, Krefeld, Germany) for 2 h at 4 °C to pellet the extracellular vesicles. This supernatant was then frozen in glass vials at −20 °C until use in ischemia-reperfusion experiments.

### 2.7. Protease Treatment of DRG-Conditioned Ischemic Solution

The DRG-conditioned ischemic solution was incubated for 15 min at 37 °C with 0.5 mg/mL protease (protease from Streptomyces griseus, Type XIV, Sigma P5147), which contains a mix of proteinase A and B and trypsin. This was performed to digest proteins in the DRG-conditioned ischemic solution, thereby eliminating larger proteins as mediators of the cardioprotective effect. To inactivate the protease, so that the protease itself does not digest the cardiomyocytes, the solution was incubated at 90 °C for 1 h, as pilot experiments indicated activity loss by this treatment. This solution was then frozen in glass vials at −20 °C until use in the ischemic phase, as described above for the DRG-conditioned ischemic solution.

### 2.8. Fractionation of DRG-Conditioned Ischemic Solution by a C18 Column

A C18 solid phase extraction column (Strata C18-E, 55 µm, 70 Å, 500 mg/6 mL, Phenomenex 8B-S001-HCH) was washed once with distilled water, and then the DRG-conditioned ischemic solution was added to the column. The flow-through and the first wash with distilled water were collected as fraction 1. Then 30%, 60%, and 100% acetonitrile were added to the column and the respective flow-throughs were collected as fractions 2–4. All solutions were driven through the column by applying pressure in the form of a syringe plunger. The solutions were then lyophilized (Martin Christ) and fraction 1 was reconstituted in water, because it contained the salts, and pH was adjusted to 6.2. The other fractions were reconstituted in the ischemic solution. These solutions were then frozen in glass vials at −20 °C until use in the ischemic phase, as described above for the DRG-conditioned ischemic solution.

### 2.9. Cellular Ischemia-Reperfusion Model

To mimic ischemia, the cell cultures were temporarily deprived of oxygen and glucose. This was achieved with an ‘ischemic solution’ with lactic acid but no glucose, elevated potassium concentration, pH of 6.2, and incubation in an anoxic environment in a modular incubator chamber (Embrient, San Diego, CA, USA) flushed with gas containing 95% N_2_ and 5% CO_2_ and placed at 37 °C for 2 h. This duration of mimicked ischemia was chosen for survival of about half of the cardiomyocytes compared to control conditions [9]. Alternatively, cardiomyocytes were treated with the “DRG-conditioned ischemic solution” instead of the “ischemic solution”. Further alternative treatments include the HEK293t- or 3T3 fibroblast-conditioned ischemic solution and the DRG-conditioned ischemic solution modified to not include extracellular vesicles, treated with protease or fractionated on a C18 column. The plates were then removed from the modular incubator chamber and the ‘reperfusion solution’ was added. The addition of the reperfusion solution restored the pH and glucose levels to physiological levels. The plates were then returned to the incubator at 37 °C and 5% CO_2_ for a further 30 min, during which time brightfield images were taken again. The same steps were performed as for the control, including fluid exchange. However, instead of using the modular incubator chamber, the cells were treated with the cardiomyocyte medium alone and placed in an incubator at 37 °C and 5% CO_2_ (Figure 1A).

### 2.10. Analysis of Cardiomyocyte Survival

Images taken before and after ischemia-reperfusion were analyzed for cardiomyocyte survival using Fiji [15]. Cardiomyocyte survival was then determined manually based on morphology: rounded cells were classified as dead, whereas elongated, rod-shaped cardiomyocytes were considered alive, following the guidelines of Mishra et al. for the assessment of myocardial cell death [19]. Assessment was performed blinded to experimental conditions. Cells that disappeared from the field of view were counted as lost. Consequently, the extent of damage in these missing cells remains unassessed and they were excluded from the analysis.

### 2.11. Metabolomics Analysis

For the mass spectrometry analysis, DRG-conditioned ischemic solutions were used in addition to the pure ischemic solution. Further, DRG were also incubated for the same time period (2 h) in an external solution in order to produce a DRG-conditioned external solution, which was measured in addition to a pure external solution. After the preparation of samples, they were immediately lyophilized and kept at −20 °C until the day of the analysis. On that day, samples were resuspended in 500 µL 80% methanol at −20 °C, incubated for 1 h at −20 °C, and centrifuged for 10 min at 4 °C on maximum speed. After that, samples were measured with LC-MS. The liquid chromatography system EXIONLC AD SYSTEM (AB Sciex, Darmstadt, Germany) was coupled to an ESI X500 QTOF mass spectrometer (AB Sciex, Darmstadt, Germany). The separation was performed using a binary gradient on a reversed phase C18 column (Kinetex; 2.1 mm × 10 cm, 2.6 μm, 100 Å, Phenomenex, Aschaffenburg, Germany). The mobile phases included mobile phase A (Water/Formic Acid, 100:0.02) and mobile phase B (Acetonitrile/Formic Acid, 100:0.02). Gradient details: 0–0.5 min, 2% mobile phase B; 0.5–2.5 min, 2–75% mobile phase B; 2.6–4 min, 95% mobile phase B; 4–5 min re-equilibration with 2% mobile phase B; flow rate = 400 μL/min, column temperature = 40 °C, and injection volume = 5 µL of each sample. Mass spectrometric detection details: Heater temperature = 500 °C, ion source gas 1 = 30 psi, ion source gas 2 = 30 psi, curtain gas = 45 psi, spray voltages = −4.5 kV/+5.0 kV for negative/positive ion mode ionization, respectively; TOFMS and TOF/MSMS scan ranges = 50–1500 *m*/*z*, accumulation time = 0.1 ms, decluttering potential = −80 V, collision energy = 45 V, and voltage spread = 15 V. Lipids were identified using MSDIAL ver.4.9.221218 Windowsx64 [20] via the integrated Lipidblast package [21]; the identification of the lipids was pursued in both negative and positive modes, and we ensured choosing the right modifier type in the MSDIAL Lipidblast MSP file tab and also selected all the possible adducts available in the adduct tab. Metabolites were identified using MSDIAL ver.4.9.221218 Windowsx64 via the spectral database package “ESI(+)-MS/MS from standards+bio+in silico (16,995 unique compounds), last edit 21 August 2022”. When peak annotation was not possible using the included spectra library, we used HMDB [22] and METLIN Gen235 [23] (purchased 20 January 2023). A volcano plot was generated using Metabolanalyst [24,25].

### 2.12. Statistical Analyses

To analyze the survival of cardiomyocytes in the cellular model of ischemia-reperfusion, a generalized linear mixed model with a binomial target distribution and a logit link was used. These models correspond to binary logistic regression, where the unit of observation was a cardiomyocyte, and its survival status (whether it survived or not) served as the binary target variable. On every experimental day, cardiomyocytes from one mouse were isolated and distributed across wells. The concept of ‘experimental day’ was used as a random factor in all models, as observations from the same day were related. Several images were acquired from each well on each experimental day, and cell counts within each image were noted. The fixed factors ‘well’ and ‘image’ were added to the statistical models and used in a factorial manner with interaction terms. *p*-Values ≤ 0.05 were considered statistically significant, and there was no adjustment for multiple testing. All tests were two-sided. The statistical analysis used IBM SPSS Statistics 28 (Armonk, NY, USA). Graphs were generated using GraphPad Prism 9 (Graphpad Software, Inc, San Diego, CA, USA) and arranged in CorelDraw 2020 (Corel Corporation, Ottawa, ON, Canada).

## 3. Results

### 3.1. A Transferable Factor from DRG Increases Cardiomyocyte Survival in Ischemia-Reperfusion

We have previously shown that a co-culture of primary murine sensory neurons derived from dorsal root ganglia (DRG) could significantly improve the survival of primary murine cardiomyocytes in ischemia-reperfusion conditions [9]. In this ischemia-reperfusion protocol, DRG neurons remain undamaged [9]. However, the mechanism underlying the observed protective effect remained elusive, although the following experiments narrow down potential explanations. With regards to sensory neurons, a promising candidate was a calcitonin gene-related peptide. Upon the activation of sensory neurons, it is released and it exerts protective effects on the cardiovascular system [26]. However, the addition of the calcitonin gene-related peptide receptor antagonist olcegepant (100 nM) to the ischemia-reperfusion model did not modify the positive impact of sensory neuron co-culture on cardiomyocyte survival (Appendix A). Thus, calcitonin gene-related peptide (CGRP) as a DRG-released protective mediator was excluded.

Next, botulinum toxin A (BoNTA at 2.5 nM) was added. A reduction in the protective effect by BoNTA would have suggested released vesicles to be involved in the cardioprotective effect. However, botulinum toxin A, added to sensory neurons 24 h prior to establishment of the co-culture, did not diminish the protective effects of sensory neurons; there was even a trend towards an increase in protective effects of sensory neurons on cardiomyocytes (Appendix A), excluding release of protective factors by vesicles.

After excluding the a priori most likely currently known mediators of the protective effects, an exploratory approach was chosen. To this end, the levels of 106 cytokines were measured in supernatants of cardiomyocyte and DRG-cardiomyocyte co-cultures in control and ischemia-reperfusion conditions in an exploratory manner. This resulted in four cytokines that were at least two-fold upregulated and 10 that were at least two-fold downregulated in ischemia-reperfusion conditions and the presence of sensory neurons (Appendix A).

The experimental design of the co-culture used so far did not allow us to distinguish between effects of transferable factors and effects mediated by direct cell–cell contacts. To test whether protective factors are transferable and thus independent of cell–cell contacts, a DRG-conditioned ischemic solution was produced. Thereby, whole DRG were subjected to ischemia for 2 h. The conditioned supernatant was then transferred to cardiomyocytes during the ischemia-reperfusion model (Figure 1A). Ischemia-reperfusion led to a reduction in the estimated cardiomyocyte survival probability by 62% compared to the control (*p* < 0.001). Treatment with the DRG-conditioned ischemic solution could reverse this effect and rescued 71% of the ischemia-reperfusion damage (*p* < 0.001, vs. ischemia and vs. control, Figure 1B and Appendix A). The dilution of the DRG-conditioned ischemic solution by a factor of 10 or more leads to the loss of its cardioprotective effect (Figure 1C and Appendix A).

### 3.2. A Hydrophilic Substance from DRG Improves Cardiomyocyte Survival

The improvement in cardiomyocyte survival mediated by treatment with the DRG-conditioned ischemic solution is not mediated by extracellular vesicles, as removal of extracellular vesicles by ultracentrifugation did not reduce the improvement in cardiomyocyte survival (*p* = 0.21, Figure 2A and Appendix A). Exposure of the DRG-conditioned ischemic solution to proteases also did not reduce the cardioprotective effect (*p* = 0.65, Figure 2B and Appendix A). Since this mixture of proteases can reasonably be assumed to degrade most proteins, a protein mediator for the protective effect seems less likely. To further narrow down the mediator(s), the DRG-conditioned ischemic solution was fractionated on a C18 column. Hydrophilic fraction 1 eluted with water maintained most of the cardioprotective effect of the DRG-conditioned ischemic solution (*p* = 0.788 vs. DRG-conditioned ischemia, Figure 2C and Appendix A). In contrast, the more hydrophobic fractions, 2 to 4, eluted with 30%, 60%, and 100% acetonitrile, did not improve cardiomyocyte survival to the same extent (*p* = 0.001, *p* = 0.054, and *p* < 0.001, respectively, vs. DRG-conditioned ischemia, Figure 2C and Appendix A).

### 3.3. HEK293t Cells and 3T3 Fibroblasts Improve Cardiomyocyte Survival in Ischemia-Reperfusion

To investigate whether the observed effects are specific to DRG, we treated cardiomyocytes with the ischemic solution that was conditioned by the cell lines HEK293t or 3T3 fibroblasts. In these experiments, 49% and 36% of the ischemia-reperfusion damage could be inhibited (*p* < 0.001 for HEK293t, *p* = 0.027 for 3T3 fibroblasts, Figure 3A and Appendix A). As for DRG neurons, the dilution of the HEK293t-conditioned ischemic solution by a factor of 10 or more led to the loss of its cardioprotective effect (Figure 3B and Appendix A).

### 3.4. Metabolomic Analysis of Conditioned Solutions

To further explore potential protective substances from DRG, a mass spectrometry omics approach was employed to analyze the DRG-conditioned solutions under ischemic or control conditions. Both positive and negative ion modes were used to detect polar and nonpolar compounds influencing cardiomyocyte survival. A total of 681 metabolites and lipids were annotated or identified. Among these, 222 were nominally two-fold higher than the control (Appendix A); significantly upregulated were 38 and downregulated were 11 of these in the DRG-conditioned ischemic solution compared to the DRG-conditioned external solution (Figure 4A). The annotations were classified into lipid families, with phosphatidylethanolamines, phosphatidylcholines, and acylcarnitines being the most prevalent. Other lipid families included lysophosphatidylethanolamines, diacylglycerols, phosphatidylinositol, sphingomyelin, and ceramides (Figure 4B). Among polar compounds, only four were identified. Notably, nicotinamide and hypoxanthine were downregulated in the DRG-conditioned ischemic solution, while adenosine, inosine, and spermidine exhibited upregulation when compared to the DRG-conditioned external solution. While many of those metabolites and lipids were successfully annotated, around 234 remain unknown, 15 of which were significantly regulated (Appendix A). Thus, out of a total of 915 metabolites, 64 showed more than a two-fold change.

## 4. Discussion

In a newly established in vitro model of ischemia, transfer of the supernatant of cells exposed to an ischemia model conferred the protection of isolated murine cardiomyocytes. The respective factor(s) were not identified, but could be narrowed down by excluding extracellular vesicles and sensitivity to protease, leading to a metabolomic profiling. The latter left more options than what could be explored using this model.

### 4.1. Investigation of Neuronal Contributions in Co-Culture with Cardiomyocytes

Previously, we could show that a co-culture with primary murine sensory neurons improves cardiomyocyte survival in ischemia-reperfusion conditions [9]. This raised the question of whether the effect was due to direct interaction between cells or mediated by transferable factors. However, before pursuing this question with further experiments, the most obvious targets or factors that could confer this cardioprotection were tested using the ischemia-reperfusion co-culture model.

First, calcitonin gene-related peptide was investigated, as this neuropeptide is released by sensory neurons upon the activation of TRPA1 and TRPV1, which are sensors of noxious compounds including reactive oxygen species and acidosis, that occurs during ischemia. In addition, calcitonin gene-related peptide has been described to have cardioprotective effects in vivo [26,27]. However, blocking of the calcitonin gene-related peptide receptor with olcegepant did not reduce the cardioprotective effect of co-culturing cardiomyocytes with sensory neurons. Second, botulinum toxin A was used to block all vesicular exocytosis in sensory neurons, which also did not reduce their protective effect [28,29]. Therefore, the results indicated that the responsible substances are not secreted by classical exocytosis in sensory neurons [30,31].

The next step was to investigate whether cytokines could be responsible for the increase in cardiomyocyte survival in ischemia-reperfusion. The cardiomyocyte medium, the ischemic and reperfusion solution, and peptigel used for coating do not contain cytokines; therefore, cardiomyocytes, neurons, or both cell types are the source of cytokines in the co-culture supernatants. The amount of differentially abundant cytokines was higher with the addition of sensory neurons than with ischemia-reperfusion, which could be a sign of general cell–cell communication between the two cell types. Most of the cytokines found to be differentially regulated between single and co-culture in ischemia-reperfusion were already known to play a role in myocardial infarction. For example, osteopontin is known for its role in myocardial remodeling post myocardial infarction [32]. Osteopontin is expressed in sensory neurons [33] and could be released under ischemia-reperfusion conditions, increasing the likelihood of cardiomyocyte survival. Using the ExpressAnalyst tool, it was observed that ischemia activated the Jak-STAT (Janus kinase–signal transducer and activator of transcription) pathway, which could be suppressed by the addition of sensory neurons in control and ischemia conditions. The JAK-STAT pathway is involved in myocardial infarction and the cardioprotective mechanisms of pre-conditioning [34]. Nevertheless, further studies are warranted to clarify if JAK-Stat or other pathways are activated by the identified factors.

As these hypothesis-driven experiments did not point to a specific factor or target, these experiments motivated us to test whether the cardioprotective effect was mediated by a transferable factor and to characterize it.

### 4.2. Cardioprotective Transferable Factor(s)

These experiments showed that the protective effect that sensory neurons have on cardiomyocyte survival during ischemia-reperfusion could indeed be transferred via a conditioned ischemic solution, suggesting that transferable factors are the cause of the cardioprotective effect of sensory neurons. This rules out other means of signaling, such as cell–cell contacts. The latter was considered an option, but pilot experiments did not suggest that the rate or direction of axonal growth differed in the presence or absence of cardiomyocytes. A dilution series of the DRG-conditioned ischemic solution indicated a low abundance of the cardioprotective factors, given that the effect was lost after ten-fold dilution. The DRG-conditioned ischemic solution mediating the cardioprotective effect was characterized further. First, it was shown that extracellular vesicles were not responsible, as their removal by ultracentrifugation did not reduce the protective effect. With the used ultracentrifugation parameters of 120,000× *g* for 2 h at 4 °C, extracellular vehicles of a size above 40 nm are largely eliminated, which includes exosomes up to 150 nm [35]. However, aggregated proteins and nucleic acids might also partially be pelleted.

Further, the factor(s) in question cannot be degraded by a protease, indicating that it is not a large protein. The DRG-conditioned ischemic solution was incubated with a mixture of proteinase A and B and trypsin for 15 min at 37 °C, followed by inactivation at 90 °C for 1 h. The time and enzyme activity for digestion should be sufficient for several-fold of the maximum protein content. To avoid cell damage from residual protease activity, this had to be inactivated. The inactivation protocol has been tested in pilot experiments, as evidenced by its ability to stop protease-mediated cell detachment.

In addition, the separation of the DRG-conditioned ischemic solution on a C18 column was used as an indicator of lipophilicity of the cardioprotective factors. Hydrophilic mediators appeared to be primarily responsible for the cardioprotective effects. This initial characterization led to the question of whether this cardioprotective effect was specific to sensory neurons. Therefore, conditioned ischemic solutions were also prepared using HEK293t cells as an unrelated cell type and 3T3 fibroblasts to account for the fibroblast population in the heart. Both cell types mediated a cardioprotective effect, but to a lesser extent. However, it is unclear whether the effect is mediated by the same factor(s) or whether there are cell-type-specific factors released. Nevertheless, the other cell types suggest that pancellular mechanisms may contribute to the cardioprotective effect, which motivated a metabolomic analysis.

### 4.3. Mass Spectrometry Metabolomic Analysis

To gain insight into factors within the DRG-conditioned ischemic solution, a mass spectrometry omics analysis was performed comparing the conditioned ischemic solution with the conditioned external solution and their respective pure counterparts. Compared to the DRG-conditioned external solution, in the DRG-conditioned ischemic solution, 38 metabolites and lipids were upregulated and 11 downregulated, indicating a clear effect of ischemia.

For some of the downregulated [36] and upregulated [37] identified polar compounds, a cardioprotective effect has been described before. Among the polar metabolites, nicotinamide and hypoxanthine were downregulated in the DRG-conditioned ischemic solution, whereas adenosine, inosine, and spermidine were upregulated compared to the DRG-conditioned external solution.

Nicotinamide has been described to exert protective effects on cardiomyocytes in vitro in the context of hypoxia. Its supplementation in vitro reduced markers for apoptosis and reduced mitochondrial stress [38,39]. As nicotinamide was downregulated in the DRG-conditioned ischemic solution, it cannot be responsible for the protective effect.

The concentration of hypoxanthine was reduced in the supernatant of DRG exposed to hypoxia compared to the supernatant of non-exposed ones. Under conditions of hypoxia, ATP cannot be reproduced to a sufficient extent, which is why ADP can be further broken down to AMP, then to adenosine, inosine, and hypoxanthine. Consequently, one would have expected a concentration increase due to hypoxia. In other studies, this was indeed observed in cardiomyocytes exposed to ischemia [40]. Opposing reactions of cardiomyocytes and sensory neurons regarding hypoxanthine release might be involved in the protective role of sensory neurons.

Considering that both adenosine and hypoxanthine are generated when ADP is further broken down, it is surprising that in contrast to hypoxanthine, adenosine as well as inosine were found to be higher-concentrated in the DRG-conditioned ischemic solution compared to the DRG-conditioned external solution. Mechanistic studies support a protective effect of adenosine [41]; clinical studies have been implemented, however, and whether adenosine administration confers relevant beneficial effects is not finally answered [42,43]. Even if exogenously administered adenosine ultimately turns out not to be beneficial, this would not exclude a relevant protective effect in vivo. Systemically administered adenosine has a half-life in the range of seconds and blocks the transmission in the atrioventricular node. However, adenosine produced endogenously in the area of myocardial hypoxia would primarily have a local and prolonged effect due to continuous generation. Such locally acting adenosine might be protective. Further, adenosine formation is likely a common response of many cell types to ATP deficiency caused by hypoxia. Adenosine as a mediator would therefore be conceivable as a joint explanatory model for the observed effect of DRG, HEK293t, and 3T3 cells. Considering the above-described degradation of ATP, the observed increase in inosine is also not surprising. However, despite some preclinical trials suggesting protective effects [44,45], and some studies suggesting inosine as a biomarker for cardiac ischemia [46], the body of evidence supporting inosine as a protective molecule is limited.

We also identified some less polar metabolites, among which the most common lipid families were phosphatidylethanolamines, phosphatidylcholines, and acylcarnitines. In patients treated for myocardial infarction, lipids were the largest class of metabolites that changed in response to reperfusion, consistent with our cellular model [47]. It was described that over half the heart’s energy comes from free fatty acids, which could indicate that the release of lipids from sensory neurons can support energy production in cardiomyocytes during ischemia [48]. However, due to the lack of oxygen during ischemia, use of these fatty acids for energy generation is impaired [49]. In another study, it was shown that patients with myocardial infarction showed major changes in the lipid classes acylcarnitine and LPC in response to the ischemia. In addition, the lipid classes triglycerides, phosphatidylinositols, and lysophosphatidylcholines are associated with lower troponin levels and therefore with less myocardial damage and also with a lower probability of major adverse cardiovascular events. Both may mediate cardioprotective effects, and several mediators from these groups were detected in the DRG-conditioned ischemic solution [50]. These metabolites should be considered in light of the metabolic situation in human myocardial ischemia [47,51].

However, it is a limitation of this study that no substance or combination of substances was shown to mediate cardioprotection. The assay is time-consuming and allows for only a limited number of tests, making it suitable for the verification but not for screening of many compound combinations. Some attempts have been made in our assay, but none with sufficient replication to allow for reporting or conclusions. Rather, the conclusion is that a different approach is needed to identify the cardioprotective factor(s) and the assay described here might be used for verification. When interpreting the results, it should also be taken into consideration that this study does not constitute an animal experiment but consists of cell culture experiments. As such, no formal sample size calculation was performed. The uncertainty of the results should be interpreted based on the presented 95% confidence intervals, and whether the results also apply to intact organisms needs to be addressed in appropriate animal experiments.

## 5. Conclusions

Overall, during ischemia, sensory neurons secrete one or more cardioprotective substances that can improve cardiomyocyte survival in an ischemia-reperfusion model.

## Figures and Tables

**Figure 1 biomedicines-12-01856-f001:**
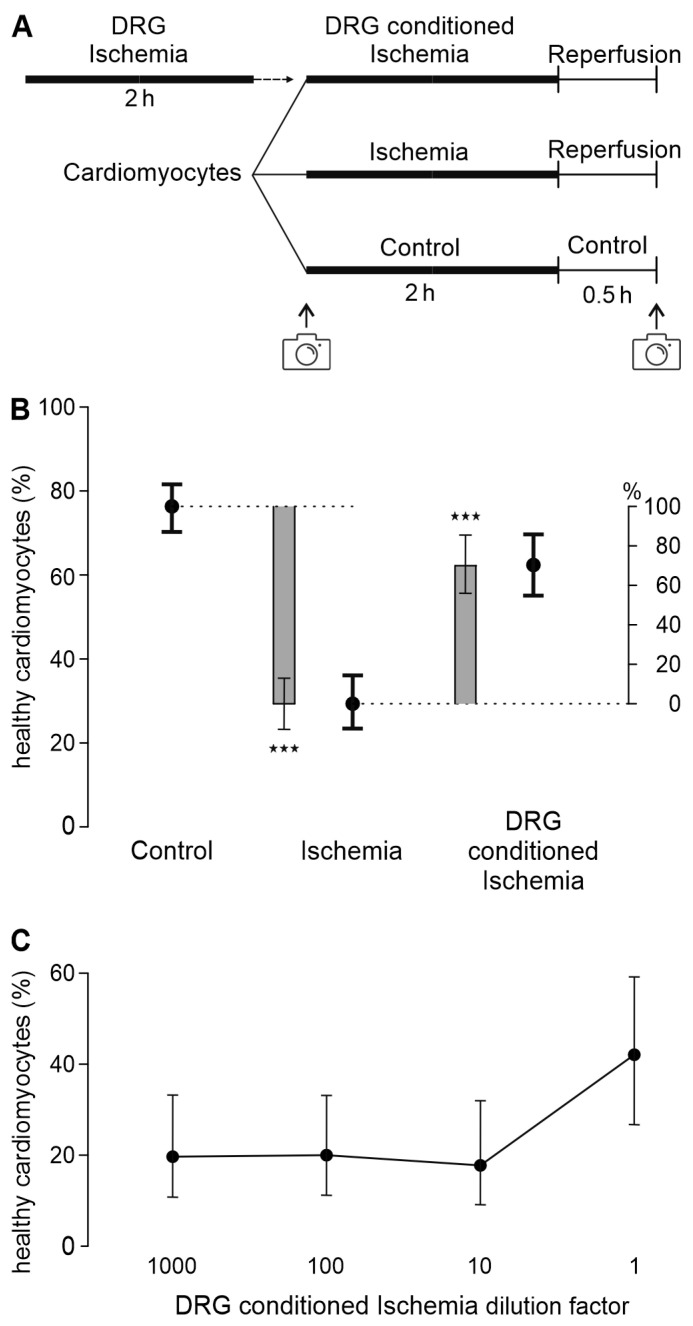
DRG exposed to ischemia release transferable mediators protecting cardiomyocytes from ischemia-reperfusion injury. (**A**) Experimental protocol of transfer of DRG-conditioned ischemic solution to cardiomyocytes in in vitro ischemia-reperfusion model. (**B**) Estimated survival probabilities for control, ischemia-reperfusion conditions, and treatment with DRG-conditioned ischemic solution are shown as black dots ± 95% CI. Estimated differences are shown as gray bars ± 95% CI. Cardiomyocytes and DRG were each derived from *n* = 24 animals, respectively. The dotted lines serves as visual reference. Cardiomyocytes of each animal were distributed to three experimental conditions, and DRG of animal sacrificed before were used to prepare conditioned ischemic solution. Data of all respective experiments are plotted in Appendix A, ★★★ *p* < 0.001. (**C**) For all dilutions of DRG-conditioned ischemic solutions, increase in cardiomyocyte survival is less than that for undiluted solution (*p* < 0.001 each). Estimated survival probabilities for each concentration are shown as black dots ± 95% CI, connected by line. Cardiomyocytes were derived from *n* = 3 animals; DRG for DRG-conditioned ischemia solution were also derived from *n* = 3 animals. Cardiomyocytes of each animal were distributed to four experimental conditions. Data separated by experimental day or per well are presented in Appendix A.

**Figure 2 biomedicines-12-01856-f002:**
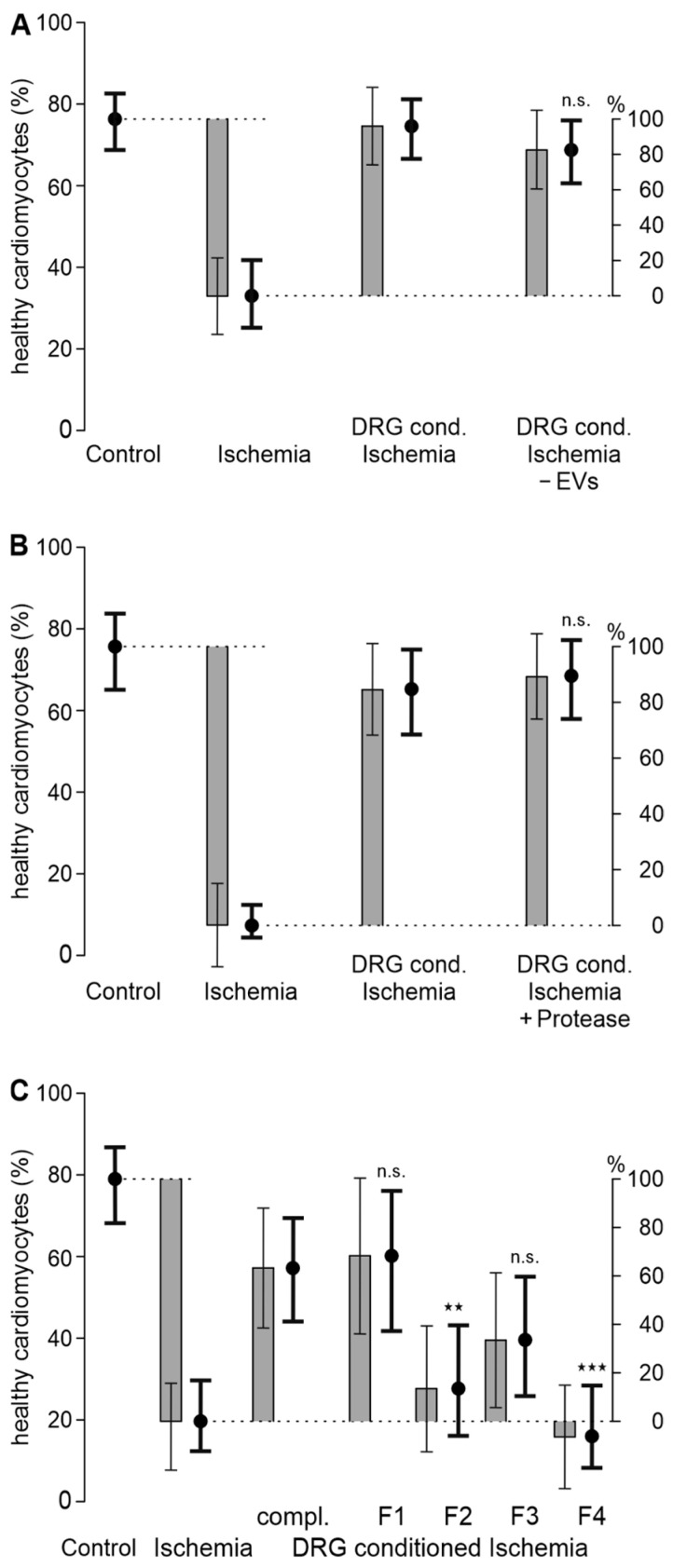
The improvement in cardiomyocyte survival by the DRG-conditioned ischemic solution is not mediated by extracellular vesicles or proteins, but rather by one or more hydrophilic mediator(s). (**A**) Cardiomyocytes were treated with the DRG-conditioned ischemic solution with and without extracellular vesicles (EVs, removed by ultracentrifugation). *p* = 0.21 vs. DRG-conditioned ischemia. Cardiomyocytes were derived from *n* = 3 animals; DRG for the DRG-conditioned ischemia solution were also derived from *n* = 3 animals. Cardiomyocytes of each animal were distributed to all 4 experimental conditions. (**B**) Cardiomyocytes were treated with the DRG-conditioned ischemic solution, and pretreated with protease or not pretreated. *p* = 0.65 vs. DRG-conditioned ischemia. Cardiomyocytes were derived from *n* = 3 animals; DRG for the DRG-conditioned ischemia solution were also derived from *n* = 3 animals. Cardiomyocytes of each animal were distributed to all 4 experimental conditions. (**C**) Cardiomyocytes were treated with the complete DRG-conditioned ischemic solution (compl.) or with 4 different fractions, F1–F4, of the DRG-conditioned ischemic solution, which were obtained using a C18 column and different eluents. Fraction 1 was eluted by water, and fractions 2–4 by 30%, 60%, and 100% acetonitrile, respectively. In all panels, estimated survival probabilities for each group are shown as black dots ± 95% CI and estimated differences are shown as gray bars ± 95% CI. The dotted lines serves as visual reference. *p*-Values refer to comparisons of respective fractions with DRG-conditioned ischemia, ★★ *p* = 0.01 and ★★★ *p* < 0.001. n.s.: not significant. Cardiomyocytes were derived from *n* = 3 animals; DRG for the DRG-conditioned ischemia solution were also derived from *n* = 3 animals. Data for each well are presented in Appendix A.

**Figure 3 biomedicines-12-01856-f003:**
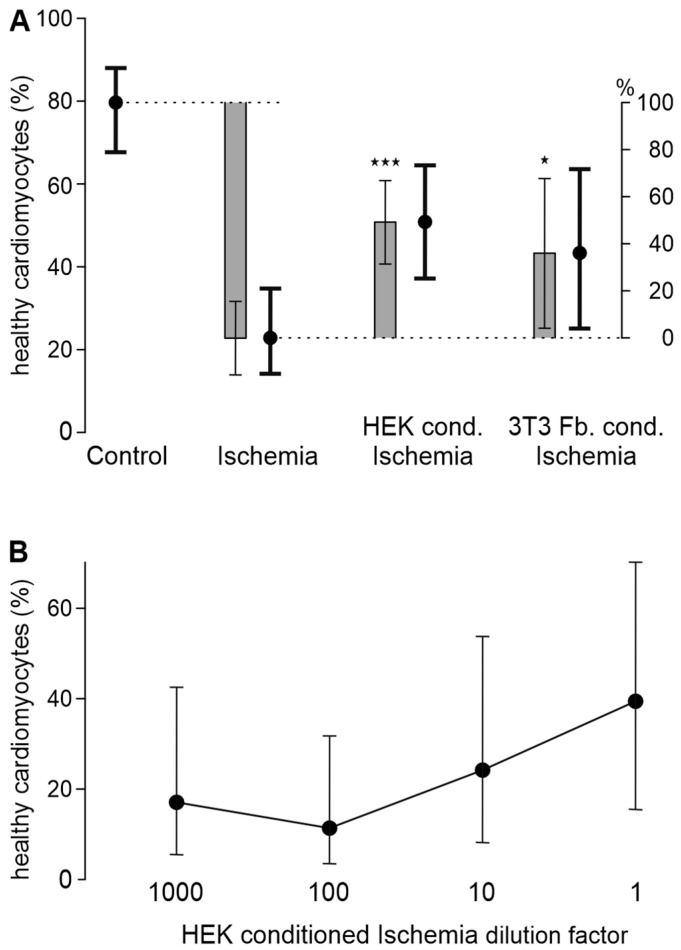
Certain cell lines can increase cardiomyocyte survival in ischemia-reperfusion. (**A**) The cell lines HEK293t and 3T3 fibroblasts (Fb.) conditioned the ischemic solution before it was applied to cardiomyocytes. Estimated survival probabilities for each cell line are shown as black dots ± 95% CI and estimated differences are shown as gray bars ± 95% CI. The dotted lines serves as visual reference. ★ *p* = 0.027 and ★★★ *p* < 0.001 vs. ischemia. Cardiomyocytes were derived from 4 to 10 animals. The exact number of animals per condition is visualized in Appendix A. (**B**) The effect of the HEK293t-conditioned ischemic solution is concentration-dependent, with dilutions by a factor of 1000, 100, and 10 shown. Estimated survival probabilities for each concentration are shown as black dots ± 95% CI, connected by a line. Data are derived from *n* = 3 animals killed on separate experimental days. Data from each experimental day are shown in Appendix A; data from each well are depicted in Appendix A.

**Figure 4 biomedicines-12-01856-f004:**
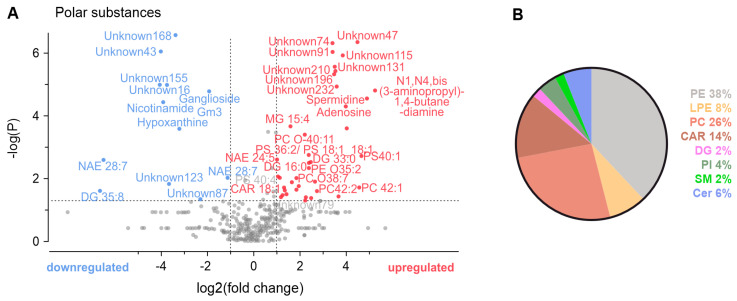
Regulated metabolites and lipids in ischemia-conditioned DRG. (**A**) A volcano plot with equal group variance showing the relative regulation in the DRG-conditioned ischemic solution compared to the reference DRG-conditioned external solution (*p*-value cut-off at 0.05, –logP(0.05) = 1.30, fold change cut-off = 2). Substances that could not be identified using reference libraries or putative annotation were labeled as ‘unknown’ and enumerated. (**B**) A pie chart of percentages of different categories of lipids found. Phosphatidylethanolamines (PEs), lysophosphatidylethanolamines (LPEs), phosphatidylcholines (PCs), acylcarnitines (CARs), diacylglycerols (DGs), phosphatidylinositol (PI), sphingomyelin (SM), and ceramides (Cer). The analysis was based on the pooled solutions generated by *n* = 9 animals per condition.

## Data Availability

The authors declare that the data supporting the findings of this study are available within the paper and its Appendix A. Should any raw data files be needed in another format, they are available from the corresponding author upon reasonable request.

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
