# Peer review of "Sensory Neurons Release Cardioprotective Factors in an In Vitro Ischemia Model"

_biomedicines, 2024, doi:10.3390/biomedicines12081856_

Round 1

Reviewer 1 Report (Previous Reviewer 1)

Comments and Suggestions for Authors

The authors responded to all previously submitted comments and made the necessary adjustments to the text of the manuscript according to them. However, upon reading the revised version of the manuscript, additional comments arose that need to be addressed before publication of the manuscript.

Major concerns:

Line 30: fractionation pointed towards a hydrophilic agent. - It is not clear why the authors write so categorically only about the hydrophilic agent, although they themselves understand and show that hydrophobic factors can also be involved

Line 32: identified 64 at least twofold changed metabolites – it is not clear where this figure comes from - the authors report on line 410 about 49 metabolites

Lines 33-35: lacking a concise conclusion, such as the one made in lines 577-578. Perhaps this phrase should be repeated in the abstract before the last sentence.

Figure 2: Improvement of cardiomyocyte survival by DRG-conditioned ischemic solution is not mediated by extracellular vesicles or proteins. – this figure title corresponds only to Figure 2A and Figure 2B. It is necessary to either supplement the title of the figure, or draw up Figure 2C as a separate figure.

Line 409: A total of 681 (Suppl. Table 1) metabolites and lipids – Suppl. Table 1 presents fewer than 681 metabolites and lipids

Minor concerns:

Line 311: CGRP - decipher or abbreviate this on line 307

Line 319: most likely mediators - it is better to write most likely currently known mediators

Line 343: CI, Cardiomyocytes

Line 345: data from each animal

Line 352: data for each well within each experimental day in figure S3C. - need a verb

Legend for Supplementary figure 4: In panels B, D and F, Estimated

Legend for Supplementary figure 5: color-shape combination represents and independent experiment

Legend for Supplementary table 2: Around 230 (15 significantly regulated) metabolites – why not write the exact number 234?

Lines 424-425: Volcano Plot with equal group variance the relative regulation in DRG-conditioned ischemic solution compared to the reference DRG-conditioned external solution – the phrase needs to be corrected

Line 431: n = 9animals – space needed

Line 517: For both downregulated(45) and upregulated(46) identified polar compounds a cardioprotective effect has been described before. – It is necessary to rephrase the sentence, since it reads as if for all the polar compounds identified in this work, a cardioprotective effect has been described before.

Comments on the Quality of English Language

Line 352: data for each well within each experimental day in figure S3C.

Lines 424-425: Volcano Plot with equal group variance the relative regulation in DRG-conditioned ischemic solution compared to the reference DRG-conditioned external solution

Author Response

Reviewer 2 Report (New Reviewer)

Comments and Suggestions for Authors

The authors investigated the role of sensory neurons response to acute and prolonged ischemia. They concluded that  during ischemia sensory neurons secreted one or more cardioprotective substances that could improve cardiomyocyte survival in an ischemia-reperfusion model. However, it is a limitation of the study that no substance or combination of substances was shown to mediate cardioprotection. The research design is absolutely appropriate.

The idea that sensory neurons could increase cardiomyocyte survival after ischemia-reperfusion injury is original and novel. The performed investiogation 

The authors investigated the role of sensory neurons during IR adequate and through several ways (i.e.DRG conditioned ischemia, DRG cond sichemia+protease, DRG cond. isch.+4 different fractions).

The report is well written with only some inaccuracies mentioned further down that need further clarification.

1, Please add a citation into the Introduction section (73-75): "As we could show previously, a combination of 2 h of such ischemic conditions and 0.5 h of mimicked reperfusion leads to reduction of cardiomyocyte survival by approximately 50%, which allows to study factors that are protective and factors that are detrimental for cardiomyocytes. "

2, Please include sample size calculations for the various endpoints. Why did you choose 3 animals/group? 

Overall, I would recommend this paper for publication, because of its novelty.

Author Response

This manuscript is a resubmission of an earlier submission. The following is a list of the peer review reports and author responses from that submission.

Round 1

Reviewer 1 Report

Comments and Suggestions for Authors

The manuscript aims to identify cardioprotective factors in an in vitro ischemia model. The authors have been working in this direction for a long time, have the necessary experience and interesting preliminary research results. The results presented in the manuscript are a continuation of previously conducted studies, they are new, relevant and potentially important for the clinic. However, a number of corrections must be made to the text of the manuscript before it can be published.

Major concerns:

Lines 74-75 – “In this study we used the same model, but modified it from direct co-culture of cardiomyocytes and sensory neurons towards transfer of conditioned solutions onto cardiomyocytes”. – however, the results and supporting materials show direct co-culture experiments of cardiomyocytes and sensory neurons (Figure 1S). - It is necessary to describe the research objectives more correctly.

Lines 80-81 – “The study aimed to identify soluble mediators of cardioprotection.” – This is not an entirely correctly formulated goal of the study, since a mass spectrometry omics approach was employed to analyze not the hydrophilic fraction 1, but the whole DRG-conditioned solutions under ischemic or control conditions.

Line 101 - Animals – There is no description of experimental animals in the “animals” subsection. It is necessary to describe which strain of mice, what age and gender, and how many mice were used in the experiments.

Line 133 – “3 identical experiments” - It’s not clear what the authors mean by 3 experiments – isolating cardiomyocytes from the hearts of three mice?

 Line 137 – “then incubated with the samples overnight “ - must be specified with which samples and how many samples and membranes were analyzed?

Line 164 - adult wild type C57/Bl6J mice - it is necessary to indicate the age and sex of the mice, as well as how many mice there were in the experimental and control groups

Lines 171-172 – “human embryonic kidney 293t cells (HEK293t) cells and 3T3 fibroblasts” - it is necessary to describe where they came from (purchased or how they were prepared)

Line 178 -  “This DRG- or HEK293t-conditioned ischemic solution” - it is necessary to explain why a similar study (dose-response) was not carried out on 3T3 fibroblasts

Line 292 - botulinum toxin A – it is necessary to add information on what dose was used

Figure 3a – shows the effects of HEK293t cells and 3T3 fibroblasts. Figure 3b shows the concentration-dependent effects of the HEK293t-conditioned ischemic solution. It is not clear why the concentration-dependent effects of the 3T3 fibroblasts-conditioned ischemic solution are not shown.

Line 363 - Why was the metabolomics analysis done not on the aqueous fraction, but on whole DRG-conditioned ischemic solutions?

The authors showed that the cardioprotective effect is associated with the water-soluble fraction. In addition, the purpose of the work was The study aimed to identify soluble mediators of сardioprotection. However, the authors discuss mainly lipid molecules. This is very illogical. It is necessary to either indicate and discuss which water-soluble molecules the authors consider to be potentially promising factors that can have a cardioprotective effect, or to adjust the goals and objectives of the study. A simple listing of some polar metabolites is not sufficient for discussion.

Minor concerns:

Line 140 - HRP - needs to be decrypted

Line 168 - Mouse DRG from a mouse

Lines 183-184 – “DRG-conditioned ischemic solution was centrifuged at 300 g for 10 min at room temperature to remove debris from cells and the supernatant was then frozen at -20 °C”. - it is better to move this phrase to line 170 (at the end of the first paragraph of the Conditioned ischemic solution paragraph)

Line 217 - EV – needs to be decrypted

Line 336 – Figure 2c (X-axis) – what is compl?

Figures S3-S5 - the figures show dots of different colors, sizes and shapes - the authors should describe what the different sizes, shapes and colors of the dots in the graphs mean.

The supplementary materials include an Excel table (Minimal Dataset), but the information presented in it is not clear. This table requires further explanation.

Reviewer 2 Report

Comments and Suggestions for Authors

The manuscript of Hoebart et al. focuses on the cardioprotective effects of sensory neurons. The topic is of interest in the cardiovascular field, however several conclusions are not adequately supported by experimental evidence and several questions arose from the experiments described.

Specific comments:

1) How sensory neurons can be identified in the intact heart? Do sensory neurons interact with other neuronal populations innervating the heart, such as sympathetic neurons?

2) What about cell-cell interactions between sensory neurons and cardiomyocytes in the intact heart? And in cell cultures? The authors have to provide images of cultured cells at baseline and after I/R experiments.

3) Are the authors sure that neuronal cells isolated from dorsal root ganglia are sensory neurons? Are you sure that there are not different contaminating neuronal populations? A phenotypic characterization of cultured sensory neurons needs to be presented.

4) Does the ischemia/reperfusion protocol lead to sensory neuron cell death? Phenotyping of cultured sensory neurons (as example cell morphology, evaluation of cell death with TUNEL assay, western blotting etc) before and after I/R is required.

5) To evaluate whether the beneficial effects on I/R cardiomyocytes is operated by factors specifically released by sensory neurons, the authors compared the effects of conditioned medium of sensory neurons vs. that of 3T3 fibroblasts and HEK293t cells. Comparison would be performed considering other cardiac neuronal populations (i.e. sympathetic neurons)

6) The effects of sensory neurons condition medium on the viability of cardiomyocytes undergone I/R need to be studied in more detail. Cell viability, structure, and key cell signaling pathways need to be considered.

7) The experiments in which the authors exposed the ‘DRG-conditioned ischemic solution to pro-teases’ needs to be better explained.

8) In general, I find that several conclusions are not well supported by presented data.

Comments on the Quality of English Language

English language is quite good